# OjaKV: Context-Aware Online Low-Rank KV Cache Compression with Oja's Rule

## Abstract

The expanding long-context capabilities of large language models are constrained by a significant memory bottleneck: the key-value (KV) cache required for autoregressive generation. This bottleneck is substantial; for instance, a Llama-3.1-8B model processing a 32K-token prompt at a batch size of 4 requires approximately 16 GB for its KV cache, a size exceeding the model's weights. While KV-cache compression via low-rank projection is a promising direction, existing methods' rely on a static, offline-learned subspace that performs poorly under data distribution shifts. To overcome these limitations, we introduce **OjaKV**, a novel framework that integrates a strategic hybrid storage policy with online subspace adaptation. First, OjaKV recognizes that not all tokens are equally important for compression; it preserves the crucial first and most recent tokens in full-rank, maintaining high-fidelity anchors for attention. Second, for the vast majority of intermediate tokens, it applies low-rank compression by incrementally adapting the projection basis using Oja's algorithm for online principal component analysis. This adaptation involves a comprehensive update during prompt prefilling and lightweight periodic updates during decoding, ensuring the subspace remains aligned with the evolving context. Crucially, our framework is fully compatible with modern attention modules like *FlashAttention*. Experiments demonstrate that OjaKV maintains or even improves zero-shot accuracy at high compression ratios. In particular, OjaKV achieves its strongest gains on very long-context benchmarks that require complex reasoning, highlighting the importance of online subspace adaptation in dynamically tracking context shifts. Furthermore, our approach is compatible with token-selection methods, enabling compounded memory savings. These results establish our hybrid framework as a practical, plug-and-play solution for memory-efficient long-context inference without requiring model fine-tuning. Code at `https://anonymous.4open.science/r/OjaKV-9D76`.

## 1 Introduction

Large language models (LLMs) such as GPT-4o (OpenAI et al., 2024) and Deepseek-R1 (DeepSeek-AI et al., 2025) have demonstrated remarkable performance across diverse domains, including coding (Nam et al., 2024), mathematics (Setlur et al., 2024), and open-ended text generation (Kumichev et al., 2024). However, as model capabilities and context length expand, GPU memory emerges as a critical bottleneck for inference. The memory footprint arises from two primary sources: (i) model weights, with a model like Llama-3.1-8B requiring 16 GB alone; and (ii) the Key-Value (KV) cache used during prompt prefilling and autoregressive decoding. For instance, processing a 32K-token prompt with Llama-3.1-8B in float16 precision at a batch size of 4 consumes an additional 16 GB for the KV cache, rivalling the size of the model weights themselves. This substantial memory consumption makes long-context inference prohibitive on all but high-end hardware.

To mitigate this challenge, a variety of methods have been proposed to optimize KV-cache memory usage (Shi et al., 2024). These approaches can be grouped into four categories: (1) *Quantization*, which stores keys and values at a lower precision (e.g., 8-bit) (Hooper et al., 2024; Liu et al., 2024); (2) *Token Selection*, which prunes or merges tokens deemed unimportant based on attention scores or heuristic saliency measures (Xiao et al., 2023; Li et al., 2024; Zhang et al., 2023); (3) *Offloading*, which transfers the KV cache to CPU memory and selectively streams it back during decoding (Tang et al., 2024; Sun et al., 2024; Zhu et al., 2025); and (4) *Low-rank Approximation*, which projects keys and values into a lower-dimensional subspace (Saxena et al., 2024; Lin et al., 2024).

Our work focuses on this fourth direction. By compressing each key and value vector from dimension $d$ (e.g., $d = 128$) to $r$ (e.g., $r = 96$), low-rank methods can reduce cache memory by $((1 - r/d) \times 100)\%$ while preserving model accuracy, yielding substantial savings for long-context inference. While token selection has become a widely adopted strategy, we provide theoretical support showing that low-rank projection and token eviction are compatible, offering multiplicative benefits that enable even greater memory reductions when combined.

Existing low-rank methods fall into two main categories. (1) *Weight-decomposition* techniques directly factorize the linear projection weights for query, key, and value ($\boldsymbol{W}_q, \boldsymbol{W}_k, \boldsymbol{W}_v$) into low-rank matrices, thereby caching already-compressed intermediate states (Chang et al., 2024). However, this approach often incurs a noticeable degradation in accuracy. (2) *Projection-based* techniques learn fixed orthonormal projeciton bases ($\boldsymbol{U}_q, \boldsymbol{U}_k, \boldsymbol{U}_v$) from a calibration dataset. These bases are then used to compress the KV cache, which is reconstructed during attention computation (Saxena et al., 2024; Lin et al., 2024). While effective, these static bases implicitly assume that inference prompts will follow the same distribution as the calibration data. In practice, distribution shifts (e.g., from dialogue to code generation) cause the approximation to deteriorate, harming generation quality.

To address these limitations, we propose **OjaKV**, a novel framework for KV-cache compression that operates on two core principles. Our first key insight is that uniform compression across all tokens is suboptimal. Motivated by the findings of attention sinks (Xiao et al., 2023), OjaKV employs a hybrid storage policy that strategically excludes the crucial first and most recent tokens from low-rank projection. This preserves their full-rank fidelity, creating stable anchors for the attention mechanism and forming a significantly stronger performance baseline. Second, for the remaining intermediate tokens, we incorporate online subspace adaptation using Oja's incremental principle component analysis (PCA) (Oja, 1997). This mechanism performs a comprehensive update during the prefill stage on a selection of salient tokens, and subsequently executes periodic lightweight updates during decoding. This ensures the low-rank basis continuously adapts to the evolving context with negligible overhead. Our framework is fully compatible with modern attention modules such as FlashAttention (Dao et al., 2022), ensuring practicality in real-world long-context inference.

We evaluate OjaKV on multiple-choice benchmarks from the lm-eval-harness (Biderman et al., 2024), aligning with prior studies (Saxena et al., 2024; Lin et al., 2024). Additionally, for the first time to the best of our knowledge, we evaluate projection-based low-rank KV cache compression methods' performance on generation-centric long-context tasks using LongBench (Bai et al., 2023) and RULER (Hsieh et al., 2024). Across all settings, OjaKV demonstrates superior performance over static low-rank baselines at the equivalent compression ratios.

In summary, our contributions are fourfold. First, we introduce a hybrid low-rank KV-cache compression framework, OjaKV, which combines a selective full-rank storage policy with a context-aware online subspace adaptation. Second, our design is compatible with FlashAttention, ensuring practicality for modern long-context inference pipelines. Third, we conduct the first comprehensive evaluation of low-rank KV compression on challenging generation-centric benchmarks, moving beyond the simpler multiple-choice tasks. Finally, we provide a theoretical analysis demonstrating the composability of OjaKV with token-eviction techniques, enabling compounded memory savings.

## 2 RELATED WORK

Recent work on reducing memory footprint of LLM inference has led to various strategies for compressing the KV cache, a primary contributor to memory overhead during long-context generation. These approaches span orthogonal directions, including quantization, token pruning, offloading, and subspace compression. Our method builds on low-rank approximation, extending it with an online, context-adaptive formulation. We review relevant methods below and highlight how OjaKV differs.

### 2.1 KV-CACHE COMPRESSION

The substantial memory overhead of the KV cache has motivated a range of compression strategies, which can be grouped into four major categories (Shi et al., 2024). (1) *Quantization*: These methods reduce memory by storing keys and values at lower precision, for example, using 4-bit integers instead of the standard 16-bit floats. KVQuant (Hooper et al., 2024) proposes a suite of techniques to

enable accurate KV cache quantization below 4-bit precision. KIVI (Liu et al., 2024) further introduces a tuning-free 2-bit scheme that quantizes keys per-channel and values per-token, achieving up to $2.6\times$ memory reduction and $2.35\text{-}3.47\times$ throughput gains with minimal quality loss. (2) *Token selection*: These methods discard or merges tokens deemed less important. StreamingLLM (Xiao et al., 2023) leverages the "attention sink" phenomenon, retaining the first and most recent tokens while discarding others. SnapKV (Li et al., 2024) heuristically selects salient tokens based on computed importance scores. (3) *Offloading*: These approaches move KV cache storage from GPU to CPU memory and selectively reload parts as needed. Quest (Tang et al., 2024) is a query-aware method that estimates the criticality of cache pages via query vectors and loads only the top-$K$ pages, thereby accelerating self-attention without accuracy loss. ShadowKV (Sun et al., 2024) offloads value caches to CPU memory and streams back only relevant chunks during decoding. (4) *Low-rank approximation*: These methods project keys and values into a lower-dimensional subspace (Saxena et al., 2024; Lin et al., 2024), either by factorizing the projection weights or applying a learned low-rank basis to construct a compressed KV cache. Our method falls into the last category and is *orthogonal* to the first three, making it complementary and enabling additive memory savings when combined.

## 2.2 Low-Rank Approximation for Attention

Low-rank structure in Transformer activations has long been leveraged to compress both weights and activations. Palu (Chang et al., 2024) and ReCalKV (Yan et al., 2025) factorize the model weights into low-rank matrices, cache compressed intermediate states, and reconstruct the full key and value tensors during attention. However, these methods often incur noticeable accuracy degradation due to lossy factorization. EigenAttention (Saxena et al., 2024) instead samples key/value activations from a calibration dataset, performs singular value decomposition (SVD), and selects orthonormal bases $U_k$ and $U_v$ that retain a target variance ratio. MatryoshkaKV (Lin et al., 2024) extends this idea by fine-tuning the low-rank projection matrices, thereby aligning the subspace more closely with downstream tasks. A key limitation of EigenAttention and MatryoshkaKV lies in their use of a static basis: once computed, the projections remain fixed throughout inference. This assumption breaks down when inference prompts diverge from the calibration distribution (e.g., shifting from conversational text to code), leading to degraded approximation and reduced generation quality. Our method address this gap by introducing online subspace adaptation. It continuously updates the projection matrices, enabling adaptive and context-aware low-rank approximation during inference.

## 2.3 Online Principal Component Analysis

When data arrives sequentially, as in the case of autoregressive decoding, rerunning SVD on the entire history is computationally infeasible. Instead, online PCA algorithms incrementally update the low-rank subspace as new samples arrive, without storing or recomputing the full covariance matrix. Classical approaches include perturbation techniques, incremental PCA, and stochastic optimization methods such as Oja's rule. Perturbation techniques update the eigen decomposition of the sample covariance by treating new observations as a low-rank perturbation (Gu & Eisenstat, 1994). These approaches are numerically accurate but computationally and memory intensive, making them unsuitable for high-dimensional or fast streaming data. Incremental SVD methods maintain a running factorization and updates it with each new batch of samples (Brand, 2002). This strikes a balance between efficiency and accuracy, and is often robust in practical settings, but can still be slower than stochastic updates for extremely large-scale problems like real-time LLM inference. Stochastic optimization methods, in contrast, directly optimize the expected variance using stochastic gradient updates, the most well-known being Oja's rule (Oja, 1997). Due to their efficiency, stochastic PCA methods are well suited to applications like KV cache compression during autoregressive decoding. Our proposed method, OjaKV, builds on this foundation. We apply Oja's rule to update the key and value subspaces during inference, enabling fast, adaptive, and context-aware low-rank approximation needing access to calibration data or fine-tuning.

## 3 Preliminaries: Low-Rank Attention

We begin by describing the standard attention mechanism and how it can be adapted for low-rank KV cache compression. For simplicity, we focus on a single attention head. Let the head dimension

be $d_h$ and the input sequence be $\boldsymbol{X} \in \mathbb{R}^{n \times d}$. Standard attention first projects the input into query, key, and value representations using projection matrices $\boldsymbol{W}_q, \boldsymbol{W}_k, \boldsymbol{W}_v \in \mathbb{R}^{d \times d_h}$:

$$\boldsymbol{Q} = \boldsymbol{X}\boldsymbol{W}_q, \qquad \boldsymbol{K} = \boldsymbol{X}\boldsymbol{W}_k, \qquad \boldsymbol{V} = \boldsymbol{X}\boldsymbol{W}_v,$$

where $\boldsymbol{Q}, \boldsymbol{K}, \boldsymbol{V} \in \mathbb{R}^{n \times d_h}$. The KV cache stores $\boldsymbol{K}$ and $\boldsymbol{V}$ for future use during decoding. The attention scores $\boldsymbol{A}$ are then computed as the scaled dot product between queries and keys, followed by a softmax operation. The output of the attention head, $\boldsymbol{O}$, is computed by applying these attention scores to value matrix $\boldsymbol{V}$: $\boldsymbol{A} = \mathrm{softmax}\left(\boldsymbol{Q}\boldsymbol{K}^T/\sqrt{d_h}\right) \in \mathbb{R}^{n \times n}, \boldsymbol{O} = \boldsymbol{A}\boldsymbol{V} \in \mathbb{R}^{n \times d_h}$.

### 3.1 Low-Rank KV-Cache Approximation

The core idea of low-rank approximation is to project $\boldsymbol{K}$ and $\boldsymbol{V}$ onto lower-dimensional subspaces. We define two orthonormal bases: $\boldsymbol{U}_k \in \mathbb{R}^{d_h \times r_k}$ for keys and queries, and $\boldsymbol{U}_v \in \mathbb{R}^{d_h \times r_v}$ for values, where $r_k, r_v \ll d_h$ are the desired ranks for compression. The bases satisfy $\boldsymbol{U}_k^{\mathsf{T}}\boldsymbol{U}_k = \boldsymbol{I}_{r_k}$ and $\boldsymbol{U}_v^{\mathsf{T}}\boldsymbol{U}_v = \boldsymbol{I}_{r_v}$. The low-rank bases $\boldsymbol{U}_k$ and $\boldsymbol{U}_v$ are initialized from a small calibration dataset (see Appendix A.2 for details).

Instead of caching the full-rank $\boldsymbol{K}$ and $\boldsymbol{V}$, we store their *compressed* representations:

$$\tilde{\boldsymbol{K}} = \boldsymbol{K}\boldsymbol{U}_k \in \mathbb{R}^{n \times r_k}, \qquad \tilde{\boldsymbol{V}} = \boldsymbol{V}\boldsymbol{U}_v \in \mathbb{R}^{n \times r_v}.$$

This reduces per-token storage requirement for the KV cache from $2d_h$ to $r_k + r_v$. If needed, the full-rank matrices can be approximately reconstructed via $\hat{\boldsymbol{K}} = \tilde{\boldsymbol{K}}\boldsymbol{U}_k^{\mathsf{T}}$ and $\hat{\boldsymbol{V}} = \tilde{\boldsymbol{V}}\boldsymbol{U}_v^{\mathsf{T}}$.

### 3.2 Efficient Attention Computation

This compressed representation allows for a more efficient attention calculation. Projecting queries into the shared query-key subspace yields $\tilde{\boldsymbol{Q}} = \boldsymbol{Q}\boldsymbol{U}_k$. The attention scores can be computed directly in the low-rank space, as the projection is mathematically equivalent to using the reconstructed matrices:

$$\tilde{\boldsymbol{Q}}\tilde{\boldsymbol{K}}^{\mathsf{T}} = (\boldsymbol{Q}\boldsymbol{U}_k)(\boldsymbol{K}\boldsymbol{U}_k)^{\mathsf{T}} = \boldsymbol{Q}\boldsymbol{U}_k\boldsymbol{U}_k^{\mathsf{T}}\boldsymbol{K}^{\mathsf{T}} = \boldsymbol{Q}\boldsymbol{U}_k(\boldsymbol{K}\boldsymbol{U}_k)^{\mathsf{T}} = \boldsymbol{Q}\boldsymbol{U}_k\tilde{\boldsymbol{K}}^{\mathsf{T}} = \boldsymbol{Q}\hat{\boldsymbol{K}}^{\mathsf{T}} \in \mathbb{R}^{n \times n}.$$

Thus, attention computed in the low-rank space is equivalent to using the reconstructed keys. The full attention operation is then performed in the low-rank space, and the final output is projected back to the original dimension: $\tilde{\boldsymbol{O}} = \mathrm{softmax}\left(\tilde{\boldsymbol{Q}}\tilde{\boldsymbol{K}}^{\mathsf{T}}/\sqrt{d_h}\right)\tilde{\boldsymbol{V}} \in \mathbb{R}^{n \times r_v}, \hat{\boldsymbol{O}} = \tilde{\boldsymbol{O}}\boldsymbol{U}_v^{\mathsf{T}} \in \mathbb{R}^{n \times d_h}$.

We note that the final output computation in the above formulation is mathematically equivalent to using the reconstructed keys and values, i.e., $\hat{\boldsymbol{O}} = \mathrm{softmax}\left(\boldsymbol{Q}\hat{\boldsymbol{K}}^{\mathsf{T}}/\sqrt{d_h}\right)\hat{\boldsymbol{V}}$.

### 3.3 Practical Implementation: Compatibility with FlashAttention

Optimized attention kernels like FlashAttention operate on full-dimensional tensors of shape $(n \times d_h)$ and cannot directly use the compressed features. To maintain compatibility, we store the compressed KV cache $(\tilde{\boldsymbol{K}}, \tilde{\boldsymbol{V}})$ and perform on-the-fly reconstruction before using FlashAttention. Given queries $\boldsymbol{Q} \in \mathbb{R}^{n \times d_h}$, we reconstruct the keys and values in the original space from the compressed cache:

$$\hat{\boldsymbol{K}} = \tilde{\boldsymbol{K}}\boldsymbol{U}_k^{\mathsf{T}} \in \mathbb{R}^{n \times d_h}, \qquad \hat{\boldsymbol{V}} = \tilde{\boldsymbol{V}}\boldsymbol{U}_v^{\mathsf{T}} \in \mathbb{R}^{n \times d_h}.$$

These reconstructed tensors are then passed to FlashAttention:

$$\boldsymbol{O}_{\mathrm{out}} = \hat{\boldsymbol{O}} = \mathrm{FlashAttention}(\boldsymbol{Q}, \hat{\boldsymbol{K}}, \hat{\boldsymbol{V}}).$$

This approach maintains the memory savings of a compressed cache while incurring only a modest runtime overhead, as detailed in Appendix A.3.

## 4 Motivation

Offline low-rank bases are fitted to the calibration distribution and can misalign under domain or task shifts at inference, increasing projection error for keys and values. We therefore maintain an adaptive

basis that periodically refreshing the basis with key and value features from current prompts and generated content the model can track distributional shifts and maintain alignment between the low-rank subspace and the evolving sequence. To validate this hypothesis, we conduct an experiment, summarized in Table 1. We compute an initial basis, $U_{\text{cal}}$, from a general-domain corpus (WikiText-2 (Guo et al., 2020)) and evaluate it on a long-context news summarization task from a different domain (the MultiNews subset of LongBench (Bai et al., 2023)). We compare this against an adapted basis, $U_{\text{adapt}}$, formed by updating $U_{\text{cal}}$ online with Oja's rule after processing a short prefix of the MultiNews data. An oracle basis, $U_{\text{test}}$, computed via PCA on the full test set, serves as an upper bound on performance.

We use two metrics: **Residual-Energy Ratio (RER)** (Najafzadeh & Mahmoudi-Rad, 2024) measures the projection error, and **Subspace Overlap (SO)** (Knyazev & Zhu, 2012), which we compute against the oracle basis ($U_{\text{test}}$) to quantify alignment, defined as $\text{SO}(U_1, U_2) = \text{Tr}(U_1^\mathsf{T} U_2 U_2^\mathsf{T} U_1)/r$. The in-domain RER of $U_{\text{cal}}$ on the calibration set is 0.035. As shown in Table 1, the static calibration basis generalizes poorly under distribution shift: its RER in-

Table 1: RER and SO on the MultiNews test set. Adapting the basis online reduces the projection error (RER) and improves alignment with the oracle test basis ($U_{\text{test}}$).

| Basis | RER (on Test)↓ | SO with $U_{\text{test}}$ ↑ |
|---|---|---|
| $U_{\text{cal}}$ | 0.255 | 0.597 |
| $U_{\text{adapt}}$ | 0.097 | 0.653 |

creases to 0.255 on the new task. Applying a lightweight Oja update to form $U_{\text{adapt}}$ mitigates this, lowering the RER to 0.097. This adaptation also improves alignment with the oracle basis, raising the SO from 0.597 to 0.653. These findings confirm that online updates can effectively counteract distribution shift.

## 5 METHODOLOGY

Our method, OjaKV, introduces a hybrid strategy for memory-efficient inference that combines selective full-rank retention, with the goal of preserving high-fidelity representations for critical tokens, with online-adapted low-rank compression for remaining sequence. As illustrated in Figure 1, most key and value vectors are projected into a compact subspace via learned projection matrices, which are continuously adapted during inference to remain aligned with the evolving context.

Our framework is built around three core components: (i) a hybrid KV cache storage policy that exempts key contextual tokens from compression; (ii) a two-phase online update scheme using Oja's algorithm to adapt the low-rank subspace during both the prompt (prefill) and decoding stages; (iii) a lightweight initialization procedure that seeds the projection matrices from a small calibration corpus (see Appendix A.2 for details).

### 5.1 KV STORAGE POLICY WITH FULL-RANK EXEMPTIONS

In long-context generation, not all tokens contribute equally to downstream predictions. Motivated by the findings of attention sinks (Xiao et al., 2023), we identify two token regions that play a critical role in shaping model outputs: (i) the first $n_{\text{start}}$ tokens of the prompt, which often contain crucial context (e.g., instructions), and (ii) the last $n_{\text{recent}}$ tokens of the prompt, which heavily influence local generation. OjaKV exempts these tokens from compression, storing their full-rank key and value vectors of dimension $d_h$. All other tokens are compressed by projecting them onto the low-rank space and stored: $\tilde{K} = KU_k, \tilde{V} = VU_v$, where $U_k, U_v \in \mathbb{R}^{d_h \times r}$ are the learned projection bases (with $r \ll d_h$). This hybrid policy preserves the most influential early and recent context at full fidelity while leveraging low-rank compression for the vast majority of the sequence.

### 5.2 TWO-PHASE ONLINE UPDATES WITH OJA'S ALGORITHM

While the hybrid storage policy preserves critical contextual anchors, the vast majority of tokens in a long sequence are still subject to low-rank compression. To ensure this representation remains accurate as the data distribution shifts during inference, we update the low-rank bases online using Oja's rule (Oja, 1982). The general form of the Oja update for a new input vector $x_t$ is:

$$y_t = U^\mathsf{T} x_t, \qquad U \leftarrow U + \eta \left( x_t - U y_t \right) y_t^\mathsf{T},$$

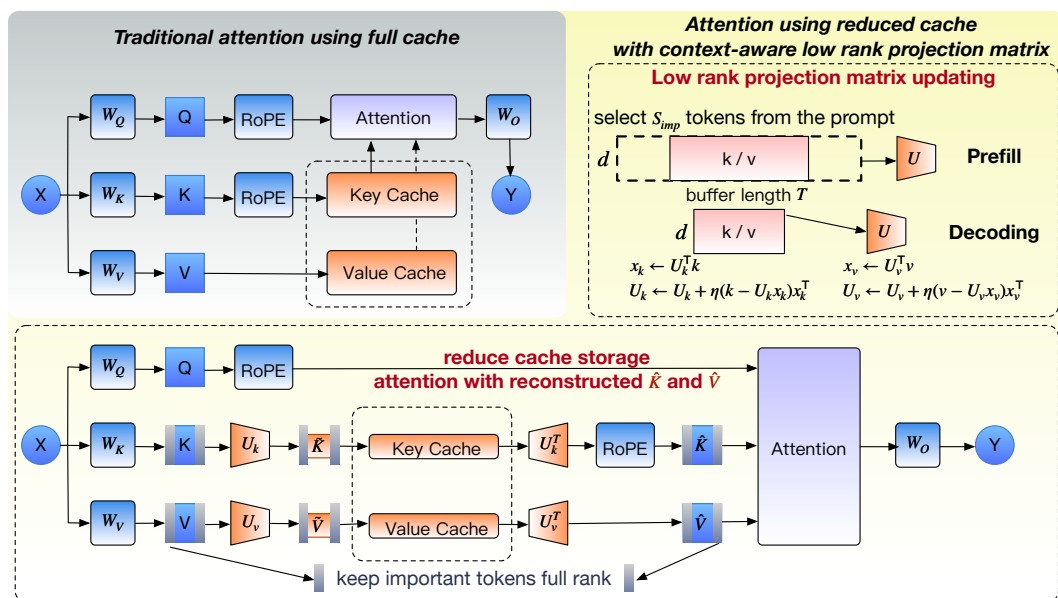

Figure 1: Overview of the OjaKV workflow. The **top-left panel** shows standard attention using full-rank KV caching. Our method, shown in the **bottom panel**, introduces a low-rank path where keys and values are compressed using projection matrices $(U_k, U_v)$ before caching. The **top-right inset** illustrates the core mechanism: these projection matrices are dynamically updated during both the prefill and decoding phases to adapt to the context.

where the input vector $x_t$ is either a key vector (when updating $U_k$) or a value vector (when updating $U_v$), $U$ is the corresponding projection basis, $y_t$ is its compressed representation, and $\eta$ is the learning rate (see Algorithm 1 in Appendix A.1 for details). This strategy is tailored to the two distinct phases of autoregressive generation: prompt processing (prefill) and token-by-token decoding.

**Prefill Stage (Prompt Processing).** During prefill, the model processes a long prompt of length $n \gg 1$. An update on every token would be counterproductive, as the inclusion of redundant information could obscure the principal directions for subspace adaptation. Instead, we select a small subset of the most salient tokens for the update using a token selection process inspired by SnapKV (Li et al., 2024).

We compute importance scores by measuring the attention from the last several query tokens to all key tokens in the prompt $(K_{\text{all}})$: $A = \text{softmax}\left(Q_{\text{last}} K_{\text{all}}^{\mathsf{T}} / \sqrt{d_h}\right)$, and define token importance as $s_t = \sum_{h,w} A_{h,w,t}$, $\mathcal{S}_{\text{imp}} = \text{TopK}(s_t)$, where $s_t$ measures how much attention token $t$ receives across all heads $(h)$ and the final window of queries $(q)$. Specifically, for each key token $t$, we compute an aggregate importance score $s_t$ by summing the attention it receives across all heads $(h)$ and all queries in the final window $(w)$. Oja's rule is then applied only to the selected set $\{x_t \mid t \in \mathcal{S}_{\text{imp}}\}$ with a prefill-specific learning rate $\eta_{\text{pre}}$. After processing all selected tokens, we perform a single QR decomposition to re-orthonormalize the updated bases to maintain numerical stability.

**Decoding Stage (Autoregressive Generation).** During decoding, a single new key-value pair $(k_t, v_t)$ is generated at each step. These vectors are temporarily stored in full-rank buffers $\mathcal{B}_k$ and $\mathcal{B}_v$. The update is performed periodically: every $T$ steps, we apply Oja's rule to all vectors accumulated in the buffers since the last update. This stage uses a smaller, more conservative learning rate $\eta_{\text{dec}} < \eta_{\text{pre}}$ to ensure stability. After each periodic update, the bases are re-orthonormalized, the buffers are cleared, and the process repeats.

This two-phase update scheme allows OjaKV to maintain alignment between the compressed subspace and the evolving context, without incurring significant computational or memory overhead. Pseudocode is provided in Appendix A.1, and a detailed case study comparing static and adaptive projections is presented in Appendix A.6.

Table 2: Retrieval accuracy (0–1; higher is better) on the RULER using LongChat-7b-v1.5-32k.

| Method | 16K | | | | | | | 32K | | | | | | |
|---|---|---|---|---|---|---|---|---|---|---|---|---|---|---|
| | S1 | S2 | MK1 | MQ | MV | QA | Avg | S1 | S2 | MK1 | MQ | MV | QA | Avg |
| FullKV | 1.00 | 0.99 | 0.91 | 0.70 | 0.71 | 0.13 | 0.74 | 0.31 | 0.67 | 0.49 | 0.44 | 0.34 | 0.02 | 0.38 |
| Eigen-N 0.8x | OOM | OOM | OOM | OOM | OOM | OOM | N/A | OOM | OOM | OOM | OOM | OOM | OOM | N/A |
| StaticPCA 0.8x | 0.68 | 0.17 | 0.23 | 0.48 | 0.26 | 0.13 | 0.33 | 0.21 | 0.07 | 0.06 | 0.04 | 0.00 | 0.08 | 0.08 |
| StaticPCA-H 0.8x | 0.97 | 0.62 | 0.49 | 0.65 | 0.61 | 0.14 | 0.58 | 0.32 | 0.31 | 0.26 | 0.17 | 0.20 | 0.08 | 0.22 |
| **OjaKV** 0.8x | **0.99** | **0.84** | **0.57** | **0.65** | **0.65** | **0.18** | **0.65** | **0.40** | **0.40** | **0.32** | **0.22** | **0.24** | **0.08** | **0.28** |
| Eigen-N 0.6x | OOM | OOM | OOM | OOM | OOM | OOM | N/A | OOM | OOM | OOM | OOM | OOM | OOM | N/A |
| StaticPCA 0.6x | 0.59 | 0.05 | 0.20 | 0.31 | 0.31 | 0.15 | 0.27 | 0.06 | 0.04 | 0.04 | 0.04 | 0.00 | 0.03 | 0.04 |
| StaticPCA-H 0.6x | 0.97 | 0.39 | 0.29 | 0.41 | 0.33 | 0.14 | 0.42 | 0.11 | 0.26 | 0.25 | 0.08 | 0.15 | 0.08 | 0.15 |
| **OjaKV** 0.6x | **1.00** | **0.65** | **0.42** | **0.44** | **0.40** | **0.16** | **0.51** | **0.23** | **0.32** | **0.34** | **0.12** | **0.16** | **0.09** | **0.21** |

## 6 EXPERIMENTS

In this section, we present the experimental setup and evaluate our proposed method, **OjaKV**. Our goal is to assess its performance against relevant baselines in realistic, long-context inference scenarios. All experiments are conducted on NVIDIA H100 GPUs and evaluated on three diverse benchmarks: **RULER** (Hsieh et al., 2024), **LongBench** (Bai et al., 2023), and **lm-eval-harness** (Agarwal et al., 2024), using **Llama-2-7B** and **Llama-3.1-8B** (Grattafiori et al., 2024) models. For a comprehensive evaluation, we compare **OjaKV** against four key baselines. We use the uncompressed **Full KV Cache** as a performance upper bound and include **Eigen-N**, a direct low-rank implementation of prior work (Saxena et al., 2024) that is impractical for long contexts due to its incompatibility with FlashAttention. We also include **StaticPCA**, which uses the same fixed, offline-computed SVD basis as Eigen-N but is adapted for modern inference by reconstructing full-rank tensors on-the-fly before the attention computation. Our primary comparative baseline is **StaticPCA-H**, which extends StaticPCA by including a token selection policy using attention sink and recent window similar to the hybrid storage design in OjaKV. We measure model accuracy, memory usage (GB), and Time to First Token (TTFT), with detailed experimental setups in Appendices A.4 and A.5.

### 6.1 RULER

**Setup.** We begin by benchmarking OjaKV on the RULER long-context retrieval suite. For our experiments, we use LongChat-7b-v1.5-32k. We evaluate its performance on challenging input sequences of both 16K and 32K tokens, pushing the model to its long-context limits and creating significant GPU memory pressure. We report results under three cache budgets: uncompressed (Full), 20% savings ($0.8\times$), and 40% savings ($0.6\times$). For each compressed budget, we compare the StaticPCA and StaticPCA-H baselines against our context-aware OjaKV, which dynamically updates its projection bases during decoding.

**Results.** On the demanding long-context tasks of RULER, the limitations of native low-rank methods become apparent. The Eigen-N baseline is not feasible in this setting, as its incompatibility with FlashAttention leads to OOM errors on sequences of this length. As reported in Table 2, OjaKV achieves strong retrieval accuracy across the RULER subtasks for both sequence lengths. It consistently outperforms the StaticPCA and StaticPCA-H baselines across all tasks and compression ratios, further validating the effectiveness of our dynamic, context-adaptive framework in extreme long-context scenarios. The other static low-rank methods perform worse since RULER tests the model's ability to perform retrieval on long, dynamic context, where there are distractors and important information constantly shifts. Here, OjaKV can better track the evolving state of the prompt, leading to enhanced performance. In these challenging scenarios, **OjaKV** effectively mitigates the performance loss from compression, while significantly using less KV cache memory.

### 6.2 LONGBENCH

We further evaluate OjaKV on LongBench, a benchmark designed to test long-context inference across diverse tasks, including single and multi-document QA, few-shot learning, and code generation. The task input lengths vary from a few thousand to over 20K tokens. As shown in Table 3, OjaKV outperforms StaticPCA-H and StaticPCA across both models and compression ratios on the majority of tasks. The performance advantage is less pronounced compared to RULER, potentially

Table 3: Accuracy (%) on LongBench tasks for Llama2-7B and Llama-3.1-8B.

| | | Single-Document QA | | Multi-Document QA | | | Few-shot Learning | | Synthetic | | Code | | Avg |
|---|---|---|---|---|---|---|---|---|---|---|---|---|---|
| | LLMs | NrtvQA | MF-en | HotpotQA | 2WikiMQA | Musique | TREC | SAMSum | PCount | PRe | Lcc | RB-P | Avg |
| Llama-2-7B | Full KV | 18.79 | 34.41 | 25.3 | 28.33 | 8.52 | 0.0 | 6.22 | 1.55 | 9.0 | 15.08 | 17.35 | 14.96 |
| | Eigen-N 0.8x | OOM | OOM | OOM | OOM | OOM | OOM | OOM | OOM | OOM | OOM | OOM | NA |
| | StaticPCA 0.8x | 16.95 | 32.8 | 21.31 | 24.73 | 6.28 | 0.0 | **5.34** | **2.25** | 2.61 | **14.06** | 16.78 | 13.01 |
| | StaticPCA-H 0.8x | 17.22 | **34.28** | 21.8 | 27.17 | 7.84 | 0.0 | 2.93 | 1.14 | 4.5 | 13.59 | **20.25** | 13.7 |
| | **OjaKV** 0.8x | **17.53** | 33.81 | **21.8** | 27.37 | 7.84 | 0.0 | 2.94 | 1.14 | **4.5** | 13.55 | 20.01 | 13.68 |
| | Eigen-N 0.6x | OOM | OOM | OOM | OOM | OOM | OOM | OOM | OOM | OOM | OOM | OOM | NA |
| | StaticPCA 0.6x | 14.21 | 27.54 | 17.41 | 19.77 | 5.11 | **0.25** | 5.65 | **3.01** | 0.5 | **13.41** | 18.84 | 11.43 |
| | StaticPCA-H 0.6x | 16.66 | 31.37 | 26.78 | 27.33 | 7.97 | 0 | 5.83 | 1.21 | 5 | 12.86 | 20.19 | 14.11 |
| | **OjaKV** 0.6x | **16.77** | **31.83** | 26.81 | 27.45 | 7.97 | 0.0 | **5.85** | 1.21 | **5.0** | 12.6 | **20.48** | **14.18** |
| Llama-3.1-8B | Full KV | 29.56 | 53.0 | 53.76 | 46.13 | 28.38 | 7.5 | 7.47 | 6.25 | 99.5 | 19.88 | 19.22 | 33.7 |
| | Eigen-N 0.8x | OOM | OOM | OOM | OOM | OOM | OOM | OOM | OOM | OOM | OOM | OOM | NA |
| | StaticPCA 0.8x | **11.81** | **49.68** | 48.06 | 43.56 | 15.43 | **10.0** | 9.46 | **3.0** | **89.5** | 21.36 | 22.46 | 29.48 |
| | StaticPCA-H 0.8x | 11.71 | 49.17 | 49.51 | 43.99 | **16.55** | 9.5 | 9.36 | 2.5 | 88 | 21.55 | **22.94** | 29.53 |
| | **OjaKV** 0.8x | 11.68 | 49.38 | **49.51** | **44.49** | 16.48 | 9.5 | **9.54** | 2.5 | 88.0 | **21.69** | 22.69 | **29.59** |
| | Eigen-N 0.6x | OOM | OOM | OOM | OOM | OOM | OOM | OOM | OOM | OOM | OOM | OOM | NA |
| | StaticPCA 0.6x | 8.16 | 39.38 | 25.63 | 22.73 | 12.65 | 0.67 | 7.1 | 1.62 | 12.5 | 18.76 | 23.48 | 15.7 |
| | StaticPCA-H 0.6x | 9.8 | **43.33** | 44.23 | **37.15** | 15.64 | 5 | **7.51** | 3.5 | 50.5 | 19.55 | **24.13** | 23.67 |
| | **OjaKV** 0.6x | **9.96** | 43.13 | **44.23** | 36.96 | **16.15** | **5.0** | 7.46 | 3.5 | 50.5 | **20.01** | 23.94 | **23.71** |

because LongBench tasks test for comprehension of a long, stable context. For Llama-2-7B, even at $0.6\times$ compression, the performance is only marginally worse on average, comapred to using the full KV cache, offering a flexible trade-off for memory savings with minimal performance drop.

## 6.3 LM-EVAL-HARNESS

To gauge the impact of KV cache compression on downstream utility, we follow the lm-eval-harness protocol on five diverse zero-shot benchmarks: *PiQA*, *WinoGrande*, *ARC-Easy*, *ARC-Challenge*, and *HellaSwag*. We evaluate all methods on the relatively short-context tasks within the lm-eval-harness, where sequence lengths are typically limited to a few hundred tokens. As shown in Table 4, we make two key observations in this setting. First, the Eigen-N and StaticPCA baselines yield identical results. This finding empirically validates our analysis in Appendix A.3, confirming the numerical equivalence between the native low-rank kernel and our FlashAttention-compatible implementation. Second, while OjaKV achieves accuracy very close to the full-rank baseline, StaticPCA-H also performs very well, suggesting that the impact of keeping a full rank attention sink and recent window, has a significant impact for these short context-tasks, as these key values contribute to a larger percentage of the total KV cache. Shorter contexts are also more robust to compression, as there is minimal accuracy drop at $0.6\times$ compression for both Llama-2-7B and Llama-3.1-8B.

Overall, our experiments on these three different benchmarks demonstrate the versatility of OjaKV, and shows that it can perform best in scenarios with long, dynamic context like in RULER.

## 6.4 EFFICIENCY

We compare OjaKV against the Full KV cache on Llama-3.1-8B-Instruct in terms of latency and GPU memory (Figure 2). The Oja update introduces some overhead to TTFT. At 32K tokens, TTFT increases from 2102 ms (Full KV) to 2801 ms (OjaKV). In contrast, memory usage, which is the limiting factor for long-context inference, decreases from 16 GB to 11.6 GB at 32K tokens. This memory reduction enables longer inputs under the same budget.

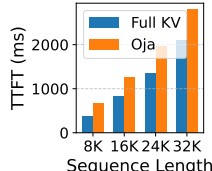
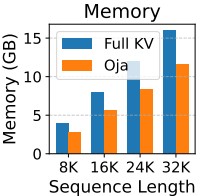

Figure 2: Efficiency comparison of Full KV and OjaKV (60% compression).

## 7 COMPATIBILITY WITH SEQUENCE LENGTH COMPRESSION

Our OjaKV method compresses the *feature dimension* $(d \rightarrow r)$ of the key and value vectors. As a result, it is orthogonal and compatible with sequence length compression techniques the *sequence length* $(n \rightarrow m)$ such as token eviction or selection. This compatibility allows their benefits to

Table 4: Accuracy (%) on lm-eval-harness tasks for Llama-2-7B and Llama-3.1-8B.

| Model | Compression Ratio | Method | Acc ↑ | | | | | |
|---|---|---|---|---|---|---|---|---|
| | | | WinoG | PiQA | HellaS | Arc-e | Arc-c | Avg-Acc |
| Llama-2-7B | Full | Baseline | 66.38 | 76.39 | 57.80 | 73.86 | 44.20 | 63.73 |
| | 0.8x | Eigen-N | 65.90 | 75.03 | 56.29 | 71.30 | 41.38 | 61.98 |
| | | StaticPCA | 65.90 | 75.03 | 56.29 | 71.30 | 41.38 | 61.98 |
| | | StaticPCA-H | 66.30 | **76.40** | **57.54** | 73.86 | 44.37 | **63.69** |
| | | **OjaKV** | **66.30** | 75.79 | 57.51 | 73.86 | **44.37** | 63.57 |
| | 0.6x | Eigen-N | 62.98 | 74.54 | 54.99 | 70.03 | 39.93 | 60.49 |
| | | StaticPCA | 62.98 | 74.54 | 54.99 | 70.03 | 39.93 | 60.49 |
| | | StaticPCA-H | 66.30 | 76.39 | 57.05 | 73.74 | 44.37 | 63.56 |
| | | **OjaKV** | **66.30** | **76.39** | **57.06** | **73.74** | **44.37** | **63.57** |
| Llama-3.1-8B | Full | Baseline | 73.88 | 80.03 | 59.04 | 81.86 | 51.88 | 69.34 |
| | 0.8x | Eigen-N | 73.56 | 79.65 | 57.74 | 81.65 | 51.54 | 68.83 |
| | | StaticPCA | 73.56 | 79.65 | 57.74 | 81.65 | 51.54 | 68.83 |
| | | StaticPCA-H | 73.95 | 79.92 | 59.14 | 81.90 | 51.79 | 69.34 |
| | | **OjaKV** | **73.95** | **79.92** | **59.14** | **81.90** | **51.79** | **69.34** |
| | 0.6x | Eigen-N | 69.85 | 78.67 | 54.87 | 79.38 | 48.12 | 66.18 |
| | | StaticPCA | 69.85 | 78.67 | 54.87 | 79.38 | 48.12 | 66.18 |
| | | StaticPCA-H | 73.95 | 79.82 | 58.35 | 81.90 | 51.88 | 69.18 |
| | | **OjaKV** | **73.95** | **79.82** | **58.35** | **81.90** | **51.88** | **69.18** |

be compounded for multiplicative savings. We briefly analyze this property theoretically here and validate it experimentally in Appendix A.7.

A token eviction policy can be represented by a selector matrix $\mathbf{S} \in \mathbb{R}^{n \times m}$. For a fixed, linear selector, our low-rank projection $\mathbf{U}^\top$ associates perfectly with the selection operation:

$$\mathbf{U}_k^\top (\mathbf{KS}) = (\mathbf{U}_k^\top \mathbf{K})\mathbf{S}, \quad \mathbf{U}_v^\top (\mathbf{VS}) = (\mathbf{U}_v^\top \mathbf{V})\mathbf{S}.$$

For advanced, context-dependent selectors where $\mathbf{S}_t = \mathrm{Sel}(\mathbf{K}, \mathbf{Q})$ is a function that selects the most relevant tokens based on the current query $\mathbf{Q}$, the commutation is not exact, but the additional projection error is bounded by the error of the selection policy itself:

$$\left\| \boldsymbol{U}^\top \boldsymbol{K} - \boldsymbol{U}^\top \boldsymbol{K} \mathbf{S}_t \right\|_F \leq \left\| \boldsymbol{K} - \boldsymbol{K}\mathbf{S}_t \right\|_F = \left\| \mathbf{K} - \mathrm{Sel}(\mathbf{K}, \mathbf{Q}) \right\|_F$$

because left-multiplication by $\boldsymbol{U}^\top$ is a contraction with respect to the Frobenius norm. This operational orthogonality means that combining a rank-$r$ OjaKV with a policy retaining $m$ of $n$ tokens yields a total compression ratio of $\mathrm{CR} = (d/r) \times (n/m)$.

## 8 CONCLUSION

In this work, we addressed the critical KV cache memory bottleneck in long-context LLMs, where static low-rank compression methods often degrade under distributional shifts. We introduced **OjaKV**, a novel framework that integrates two complementary components: a **hybrid storage policy**, which preserves critical tokens in full rank, and a lightweight, Oja-based **online update scheme** to adapt the low-rank subspace for all other tokens.

Our extensive experiments show that OjaKV consistently outperforms strong static baselines, preserving or even improving model accuracy at aggressive compression ratios. Crucially, our evaluation is one of the first to comprehensively assess low-rank methods on challenging, **generation-based** long-context tasks. Prior work has often relied on perplexity-based metrics, which we find can mask significant degradation in factual accuracy and coherence during generation. Our results reveal that while naive, uniform low-rank compression can indeed harm generation quality, OjaKV's hybrid policy effectively mitigates this issue by strategically preserving only a few key tokens in full rank. Notably, OjaKV demonstrates the largest gains on very challenging long-context benchmarks, confirming the value of online subspace adaptation in dynamically aligning the compression basis with evolving context. By ensuring full compatibility with modern inference kernels like FlashAttention and offering multiplicative savings with token-eviction methods, OjaKV establishes this hybrid approach as a practical, parameter-free paradigm for efficient long-context LLM inference.

**Future Work** A promising avenue is to replace the fixed hyperparameters in our update mechanism with dynamic schedules. Future work could explore adapting the learning rates ($\eta_{\mathrm{pre}}, \eta_{\mathrm{dec}}$) and the update buffer size ($T$) based on metrics like activation shift or generation perplexity, potentially improving both responsiveness and stability.

## USE OF LARGE LANGUAGE MODELS

LLMs were used to aid and polish the writing of this paper. Specifically, their assistance was limited to improving grammar, phrasing, and overall clarity. The authors reviewed, revised, and take full responsibility for all content, ensuring the scientific integrity of this work.

## REPRODUCIBILITY STATEMENT

We are committed to ensuring the reproducibility of our research. The complete source code is included in the anonymous repository `https://anonymous.4open.science/r/OjaKV-9D76`. Our methodology is described in the main text, with full implementation details, model configurations, and all hyperparameter settings provided in the Appendix.

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

## A APPENDIX

### A.1 ALGORITHM

We consolidate the complete OjaKV online updating process into a formal procedure in Algorithm 1. The algorithm outlines the two primary stages of our method: a comprehensive update during the prefill phase and lightweight, periodic updates during the decoding phase.

The **Prefilling Phase** is designed to adapt the initial, general-purpose projection matrices ($U_k$ and $U_v$) to the specific content of the input prompt. This process begins by identifying a small subset of the most salient tokens using an importance scoring mechanism inspired by SnapKV (Line 3). The

key and value vectors corresponding to these tokens are then used to perform a single, comprehensive batch update on the projection matrices via Oja's rule, using a relatively high learning rate, $\eta_{\text{pre}}$ (Lines 4-5). After this significant adaptation, the matrices are re-orthonormalized to maintain their geometric properties (Line 6). Finally, our hybrid storage policy is enacted by marking the critical first and last tokens of the prompt as exempt from compression (Line 7).

The **Decoding Phase** handles the continuous adaptation of the subspace as new tokens are autoregressively generated. At each step, the newly generated key-value pair is temporarily stored in a buffer (Line 11). To maintain efficiency, updates are performed periodically. Every $T$ steps, all vectors accumulated in the buffer are used for another batch Oja update, this time with a more conservative learning rate, $\eta_{\text{dec}}$, to ensure stable learning (Lines 14-15). The bases are again re-orthonormalized, and the buffers are cleared for the next cycle (Lines 16-17). This two-phase approach allows OjaKV to make a strong initial adaptation to the context while continuously refining the subspace with minimal overhead during generation.

---

**Algorithm 1** OjaKV

---

**Require:** Low rank projection matrices $\boldsymbol{U}_k, \boldsymbol{U}_v$; learning rates $\eta_{\text{pre}}, \eta_{\text{dec}}$; update buffer size $T$; exemption sizes $n_{\text{start}}, n_{\text{recent}}$; prefill importance window $w$; top-$k$ $k_{\text{pre}}$

1: **Prefilling Phase:**
2: Compute token-importance scores over the prompt using the last $w$ queries; select $\mathcal{S}_{\text{imp}} = \text{TopK}(s)$
3: Form matrices $\boldsymbol{K} = [k_t]_{t \in \mathcal{S}_{\text{imp}}}$, $\boldsymbol{V} = [v_t]_{t \in \mathcal{S}_{\text{imp}}}$
4: $\tilde{\boldsymbol{K}} \leftarrow \boldsymbol{U}_k^\mathsf{T} \boldsymbol{K}$; $\boldsymbol{U}_k \leftarrow \boldsymbol{U}_k + \eta_{\text{pre}}\left(\boldsymbol{K} - \boldsymbol{U}_k \tilde{\boldsymbol{K}}\right)\tilde{\boldsymbol{K}}^\mathsf{T}$
5: $\tilde{\boldsymbol{V}} \leftarrow \boldsymbol{U}_v^\mathsf{T} \boldsymbol{V}$; $\boldsymbol{U}_v \leftarrow \boldsymbol{U}_v + \eta_{\text{pre}}\left(\boldsymbol{V} - \boldsymbol{U}_v \tilde{\boldsymbol{V}}\right)\tilde{\boldsymbol{V}}^\mathsf{T}$
6: $(\boldsymbol{U}_k, \boldsymbol{U}_v) \leftarrow \text{Orthonormalise}(\boldsymbol{U}_k, \boldsymbol{U}_v)$
7: Mark the first $n_{\text{start}}$ prompt tokens and the last $n_{\text{recent}}$ as full-rank exempt
8:
9: **Decoding Phase:**
10: **for** step $t = 1, 2, \ldots$ **do**
11:     Generate new $(k_t, v_t)$ and append to buffers $\mathcal{B}_k, \mathcal{B}_v$
12:     **if** $t \bmod T = 0$ **then**
13:         Form $\boldsymbol{K} = [k_i]_{i \in \mathcal{B}_k}$, $\boldsymbol{V} = [v_j]_{j \in \mathcal{B}_v}$
14:         $\tilde{\boldsymbol{K}} \leftarrow \boldsymbol{U}_k^\mathsf{T} \boldsymbol{K}$; $\boldsymbol{U}_k \leftarrow \boldsymbol{U}_k + \eta_{\text{dec}}\left(\boldsymbol{K} - \boldsymbol{U}_k \tilde{\boldsymbol{K}}\right)\tilde{\boldsymbol{K}}^\mathsf{T}$
15:         $\tilde{\boldsymbol{V}} \leftarrow \boldsymbol{U}_v^\mathsf{T} \boldsymbol{V}$; $\boldsymbol{U}_v \leftarrow \boldsymbol{U}_v + \eta_{\text{dec}}\left(\boldsymbol{V} - \boldsymbol{U}_v \tilde{\boldsymbol{V}}\right)\tilde{\boldsymbol{V}}^\mathsf{T}$
16:         $(\boldsymbol{U}_k, \boldsymbol{U}_v) \leftarrow \text{Orthonormalise}(\boldsymbol{U}_k, \boldsymbol{U}_v)$
17:         Reset $\mathcal{B}_k, \mathcal{B}_v$
18:     **end if**
19: **end for**

---

## A.2   Low rank subspace initialization

We describe here the detailed procedure for constructing the initial projection bases.

For attention head $i$, we gather per-token activations of queries, keys, and values from $n_s$ sampled sequences of length $n$:

$$\mathbf{R}_i^Q = \left[(\mathbf{Q}_i^1)^\top, \ldots, (\mathbf{Q}_i^{n_s})^\top\right], \quad \mathbf{R}_i^K = \left[(\mathbf{K}_i^1)^\top, \ldots, (\mathbf{K}_i^{n_s})^\top\right], \quad \mathbf{R}_i^V = \left[(\mathbf{V}_i^1)^\top, \ldots, (\mathbf{V}_i^{n_s})^\top\right],$$

where each $\mathbf{R}_i^{(\cdot)} \in \mathbb{R}^{(n_s \cdot n) \times d_h}$ and $d_h$ is the head dimension.

To encourage a shared representation, we concatenate the query and key matrices:

$$\mathbf{R}_i^{KQ} = [\mathbf{R}_i^Q, \mathbf{R}_i^K] \in \mathbb{R}^{(n_s \cdot n) \times 2d_h}.$$

Applying compact SVD gives

$$\mathbf{R}_i^{KQ} = \boldsymbol{U}\boldsymbol{\Sigma}\boldsymbol{V}^\top,$$

with singular values $\sigma_1 \geq \cdots \geq \sigma_{d_h}$.

We select the smallest rank $r$ satisfying the energy criterion

$$\frac{\|(\mathbf{R}_i^{KQ})_r\|_F^2}{\|\mathbf{R}_i^{KQ}\|_F^2} \geq \epsilon_{\text{th}}.$$

The top-$r$ columns of $\boldsymbol{U}$ define the query–key basis $\boldsymbol{U_k} \in \mathbb{R}^{d_h \times r_k}$.

For the values, we apply the same procedure directly to $\mathbf{R}_i^V$ to obtain $\boldsymbol{U_v} \in \mathbb{R}^{d_h \times r_v}$. Finally, to maintain consistency across attention heads in a layer, we set the effective rank to the maximum $r$ observed in that layer.

## A.3 EQUIVALENCE TO FULL-RANK FLASHATTENTION AND COST COMPARISON

**Notation**  Let $\boldsymbol{Q} \in \mathbb{R}^{m \times d_h}$ and $\boldsymbol{K}, \boldsymbol{V} \in \mathbb{R}^{n \times d_h}$ denote the per-head query, key, and value blocks, where $m$ is the number of current queries and $n$ is the number of cached keys/values. At the prefilling stage, $m = n$; at the decoding stage, $m = 1$. Let $\boldsymbol{U}_k \in \mathbb{R}^{d_h \times r_k}$ and $\boldsymbol{U}_v \in \mathbb{R}^{d_h \times r_v}$ be orthonormal bases with $\boldsymbol{U}_k^\mathsf{T} \boldsymbol{U}_k = \boldsymbol{I}_{r_k}$ and $\boldsymbol{U}_v^\mathsf{T} \boldsymbol{U}_v = \boldsymbol{I}_{r_v}$. Define compressed features

$$\tilde{\boldsymbol{Q}} = \boldsymbol{Q}\boldsymbol{U}_k \in \mathbb{R}^{m \times r_k}, \quad \tilde{\boldsymbol{K}} = \boldsymbol{K}\boldsymbol{U}_k \in \mathbb{R}^{n \times r_k}, \quad \tilde{\boldsymbol{V}} = \boldsymbol{V}\boldsymbol{U}_v \in \mathbb{R}^{n \times r_v}.$$

### A.3.1 EQUIVALENCE OF TWO COMPUTATION REGIMES

We compare (a) computing attention in the reduced space and expanding the output, versus (b) reconstructing full-rank $\boldsymbol{K}, \boldsymbol{V}$ and calling a standard FlashAttention kernel.

**Low-rank kernel (compute-then-expand).**  Form reduced logits and outputs

$$\tilde{\boldsymbol{A}} = \text{softmax}\left(\frac{\tilde{\boldsymbol{Q}}\,\tilde{\boldsymbol{K}}^\mathsf{T}}{\sqrt{d_h}}\right) \in \mathbb{R}^{m \times n}, \qquad \tilde{\boldsymbol{O}} = \tilde{\boldsymbol{A}}\,\tilde{\boldsymbol{V}} \in \mathbb{R}^{m \times r_v},$$

then expand $\hat{\boldsymbol{O}} = \tilde{\boldsymbol{O}}\,\boldsymbol{U}_v^\mathsf{T} \in \mathbb{R}^{m \times d_h}$.

**FlashAttention-compatible (reconstruct-then-compute).**  Reconstruct full-rank tensors

$$\hat{\boldsymbol{K}} = \tilde{\boldsymbol{K}}\,\boldsymbol{U}_k^\mathsf{T} \in \mathbb{R}^{n \times d_h}, \qquad \hat{\boldsymbol{V}} = \tilde{\boldsymbol{V}}\,\boldsymbol{U}_v^\mathsf{T} \in \mathbb{R}^{n \times d_h},$$

and call FlashAttention with the original queries $\boldsymbol{Q}$:

$$\hat{\boldsymbol{O}} = \text{softmax}\left(\frac{\boldsymbol{Q}\,\hat{\boldsymbol{K}}^\mathsf{T}}{\sqrt{d_h}}\right) \hat{\boldsymbol{V}} \in \mathbb{R}^{m \times d_h}.$$

**Lemma (logit equivalence).**  With the above definitions,

$$\boldsymbol{Q}\,\hat{\boldsymbol{K}}^\mathsf{T} = \tilde{\boldsymbol{Q}}\,\tilde{\boldsymbol{K}}^\mathsf{T}.$$

*Proof.* Since $\hat{\boldsymbol{K}}^\mathsf{T} = (\tilde{\boldsymbol{K}}\,\boldsymbol{U}_k^\mathsf{T})^\mathsf{T} = \boldsymbol{U}_k\,\tilde{\boldsymbol{K}}^\mathsf{T}$, we have $\boldsymbol{Q}\,\hat{\boldsymbol{K}}^\mathsf{T} = \boldsymbol{Q}\,(\boldsymbol{U}_k\,\tilde{\boldsymbol{K}}^\mathsf{T}) = (\boldsymbol{Q}\,\boldsymbol{U}_k)\,\tilde{\boldsymbol{K}}^\mathsf{T} = \tilde{\boldsymbol{Q}}\,\tilde{\boldsymbol{K}}^\mathsf{T}$.

**Corollary (output equivalence).**  The two computation regimes produce the same output $\hat{\boldsymbol{O}}$.
*Proof.* Starting from the FlashAttention-compatible definition of $\hat{\boldsymbol{O}}$:

$$
\begin{aligned}
\hat{\boldsymbol{O}} &= \text{softmax}\left(\frac{\boldsymbol{Q}\,\hat{\boldsymbol{K}}^\mathsf{T}}{\sqrt{d_h}}\right) \hat{\boldsymbol{V}} && \text{(FA-compatible definition)} \\
&= \text{softmax}\left(\frac{\tilde{\boldsymbol{Q}}\,\tilde{\boldsymbol{K}}^\mathsf{T}}{\sqrt{d_h}}\right) \hat{\boldsymbol{V}} && \text{(by logit equivalence)} \\
&= \text{softmax}\left(\frac{\tilde{\boldsymbol{Q}}\,\tilde{\boldsymbol{K}}^\mathsf{T}}{\sqrt{d_h}}\right) (\tilde{\boldsymbol{V}}\,\boldsymbol{U}_v^\mathsf{T}) && \text{(substituting definition of } \hat{\boldsymbol{V}}) \\
&= \left(\text{softmax}\left(\frac{\tilde{\boldsymbol{Q}}\,\tilde{\boldsymbol{K}}^\mathsf{T}}{\sqrt{d_h}}\right) \tilde{\boldsymbol{V}}\right) \boldsymbol{U}_v^\mathsf{T} && \text{(associativity)} \\
&= \tilde{\boldsymbol{O}}\,\boldsymbol{U}_v^\mathsf{T} && \text{(low-rank kernel definition)}
\end{aligned}
$$

Hence the FlashAttention-compatible path is numerically equivalent to computing in the reduced space and then expanding, provided the same scaling $1/\sqrt{d_h}$ is used. Using $1/\sqrt{r_k}$ changes the effective temperature and usually needs calibration.

A.3.2 COMPLEXITY AND MEMORY COMPARISON

We summarize per-head costs for a single block with $m$ queries against $n$ cached keys/values. Big-O ignores softmax and masking; General matrix multiply (GEMM) shapes are shown for clarity.

| Regime | Main computations | KV memory per token |
|---|---|---|
| Full-rank baseline | $\boldsymbol{Q}\boldsymbol{K}^{\top}$: $(m \times d_h)(d_h \times n) = O(mnd_h)$ 
 $\boldsymbol{A}\boldsymbol{V}$: $(m \times n)(n \times d_h) = O(mnd_h)$ | $2d_h$ elements |
| Low-rank kernel | $\tilde{\boldsymbol{Q}}\tilde{\boldsymbol{K}}^{\top}$: $(m \times r_k)(r_k \times n) = O(mnr_k)$ 
 $\tilde{\boldsymbol{A}}\tilde{\boldsymbol{V}}$: $(m \times n)(n \times r_v) = O(mnr_v)$ 
 Expand: $(m \times r_v)(r_v \times d_h) = O(mr_vd_h)$ | $r_k + r_v$ elements |
| FA-compatible | Reconstruct $\boldsymbol{K}$: $(n \times r_k)(r_k \times d_h) = O(nr_kd_h)$ 
 Reconstruct $\boldsymbol{V}$: $(n \times r_v)(r_v \times d_h) = O(nr_vd_h)$ 
 FA kernel: $O(mnd_h)$ | $r_k + r_v$ elements |

**Discussion.** The low-rank kernel reduces the quadratic dot-product costs from $O(mnd_h)$ to $O(mnr_k)$ and $O(mnr_v)$, plus a linear expansion cost of $O(mr_vd_h)$. The FA-compatible path keeps the full-rank kernel complexity $O(mnd_h)$ but preserves memory savings by storing only $\tilde{\boldsymbol{K}}, \tilde{\boldsymbol{V}}$; the reconstruction GEMMs are linear in $n$.

**KV-cache memory in bytes.** Let $b$ be bytes per scalar (e.g., $b$=2 for float16). For $L$ layers and $H_{\mathrm{kv}}$ KV heads, the total KV memory for a sequence of length $T$ and batch size $B$ is

$$\text{Full rank:} \quad M_{\mathrm{full}} = B\,T\,L\,H_{\mathrm{kv}}\,(2d_h)\,b, \qquad \text{Low rank:} \quad M_{\mathrm{low}} = B\,T\,L\,H_{\mathrm{kv}}\,(r_k + r_v)\,b,$$

with fractional saving

$$\text{Saving} = 1 - \frac{r_k + r_v}{2d_h}.$$

When $r_k$=$r_v$=$r$, this simplifies to Saving $= 1 - \frac{r}{d_h}$.

A.4 DETAILED EXPERIMENTAL SETUP

This section provides a detailed overview of the experimental environment, models, datasets, and evaluation protocols used in this study to ensure full reproducibility of our results.

**Hardware and Software Environment.** All experiments were conducted on a single **NVIDIA H100 NVL** GPU. The software stack was built upon PyTorch and Hugging Face Transformers. The specific versions of the core libraries were as follows: **PyTorch** `torch==2.6.0`, **Transformers** `transformers==4.44.0`, and **FlashAttention** `flash_attn==2.7.4.post1`. All models were run using their standard float16 precision implementation.

**Models.** We evaluated our method on several prominent open-source Large Language Models. For clarity and reproducibility, the specific Hugging Face repository identifiers for each model were: **Llama-2-7B** (`meta-llama/Llama-2-7b-chat-hf`), **Llama-3.1-8B** (`meta-llama/Llama-3.1-8B-Instruct`), and **LongChat-7B for RULER** (`lmsys/longchat-7b-v1.5-32k`).

**Calibration Dataset.** The initial low-rank projection bases, $U_k$ and $U_v$, were derived from a small, general-domain calibration dataset. For this purpose, we used the **WikiText-2** dataset. The initialization process followed the procedure outlined in Appendix A.2, where key and value activations were collected from a number of sampled sequences and then decomposed via SVD to form the initial subspaces.

**Evaluation Benchmarks and Metrics.** Our comprehensive evaluation was performed across three diverse benchmarks: **lm-eval-harness**, **LongBench**, and **RULER**. Performance was assessed based on the following metrics. **Accuracy**: We report the specific accuracy metrics as defined by each benchmark's protocol. For lm-eval-harness, this includes the zero-shot accuracy on tasks like PiQA, WinoGrande, and HellaSwag. For LongBench and RULER, this corresponds to their respective scoring mechanisms for long-context reasoning and retrieval tasks. **GPU Memory**: Memory consumption is reported in Gigabytes (GB) and reflects the specific GPU memory allocated to KV

cache during the inference process for a given context length. This provides a practical measure of the hardware requirements. **Latency**: Latency is reported as Time To First Token (TTFT) in milliseconds (ms), which primarily measures the overhead during the prompt processing (prefill) stage. This is a critical metric for user-facing applications where initial response time is important.

## A.5 DEFAULT HYPERPARAMETERS

Table 5: Default hyperparameters unless stated otherwise.

| Symbol | Default | Description |
|--------|---------|-------------|
| $\eta_{pre}$ | 0.10 | Oja update lr during prefill |
| $\eta_{dec}$ | 0.05 | Oja update lr during decode |
| $T$ | 32 | decode update period (steps) |
| $n$ | – | prompt length |
| $k_{pre}$ | $0.05n$ | top-$k$ salient tokens at prefill |
| $w$ | 32 | importance window size (queries) |
| $n_{start}$ | 32 | full-rank exemption at the beginning |
| $n_{recent}$ | 32 | full-rank exemption at the end |

## A.6 QUALITATIVE ANALYSIS AND CASE STUDIES

To complement the quantitative results presented in the main paper, this section provides a qualitative case study. Our goal is to illustrate the practical impact of OjaKV's online subspace adaptation on generation quality, particularly in long-context scenarios where the distribution of activations can shift significantly. We use a long-document summarization task from the MultiNews subset of the LongBench benchmark, comparing summaries generated by our full OjaKV framework against the StaticPCA baseline at an aggressive 0.6x compression ratio.

**Input Document Snippet (Conceptual Summary)**

The input is a lengthy (over. 8k tokens) article. It is a multi-doc summarization that requires summarizing over multiple news.

**Reference Summary**

The Occupy LA encampment got a stay of execution this morning—though a few arrests have been reported beyond the boundaries of City Hall Park. Two hours after protesters defied city hall's order to vacate the park by 12:01am, lawyer and Occupy supporter Jim Lafferty told Reuters he expected police to give the camp a two-day reprieve. At around 5am, police told protesters over a loudspeaker that they didn't intend to clear the park, the LA Times reports. Police do, however, intend to clear protesters out of the street, and several people who refused to move out of an intersection have already been arrested. We definitely need to get morning traffic in,ä police commander explained. Everything also seems calm at Occupy Philadelphia, where protesters defied a 5pm Sunday deadline to disperse. But as of last night, a police inspector said they didn't intend to forcibly evict anyone. We look forward to working with Occupy Philadelphia and a resolution of the problem,ḧe told the Philadelphia Inquirer. Confrontation is never good.

**Baseline: StaticPCA (0.6x Compression)**

The Occupy Wall Street movement has been making headlines in various cities across the United States. In Philadelphia, Occupy Philly protesters remained on site at City Hall into the evening of November 28, despite a deadline set by Mayor Nutter for them to leave. The protesters, who had set up camp on the City Hall lawn, were given a deadline to vacate the

> area, but they refused to leave. The police had no plans to evict anyone, and the protesters were allowed to stay on the lawn.

**Analysis of Baseline:** The summary generated by StaticPCA exhibits a classic failure mode in long-context summarization: **topical fixation and critical information loss.** The source document clearly discusses events in two distinct locations, with a primary focus on the tense standoff in Los Angeles and a secondary mention of the calm situation in Philadelphia. However, the StaticPCA summary focuses *exclusively* on Philadelphia, completely omitting any mention of the events in Los Angeles, which constituted the main subject of the report. This suggests that the static principal components, established early, failed to represent the new, dominant information introduced later. The model's inability to adapt to this semantic shift resulted in a factually incomplete and highly misleading summary.

> **OjaKV (0.6x Compression)**
>
> The Occupy Wall Street movement continues to gain momentum across the United States, with protests taking place in various cities. In Los Angeles, the Occupy LA encampment at City Hall Park was the site of a tense standoff between protesters and police, with commanders later announcing they would not clear the encampment. In Philadelphia, the Occupy Philadelphia campers also remained in place despite a 5 p.m. deadline to vacate the area. The movement is a national one protesting economic inequality, and the response from law enforcement has varied by city.

**Analysis of OjaKV:** In stark contrast, the summary from OjaKV successfully captures the multi-faceted nature of the source document. It correctly identifies and synthesizes the key events from **both Los Angeles and Philadelphia**, presenting a coherent and comprehensive overview. This demonstrates the effectiveness of OjaKV's online subspace adaptation. As the model processed the document and encountered new information related to the LA protest, it dynamically updated its KV cache's principal components. This adaptation allowed it to preserve the crucial details from different sections of the long-context input, avoiding the catastrophic information loss seen in the StaticPCA baseline. The resulting summary is significantly more accurate and useful.

## A.7 EXPERIMENTAL VALIDATION OF COMPATIBILITY WITH TOKEN SELECTION

In Section 7, we posited that OjaKV, which compresses the feature dimension ($d \rightarrow r$), is orthogonal to token eviction techniques that compress the sequence length ($n \rightarrow m$). We argued that this orthogonality allows for compounded, multiplicative memory savings. This section provides empirical validation for this claim by combining OjaKV with SnapKV (Li et al., 2024), a representative token selection method.

Table 6: Compounded KV cache compression by combining OjaKV with SnapKV. The total compression ratio demonstrates multiplicative savings, offering a compelling trade-off between performance and memory efficiency.

| Method | Rank Comp. | Token Keep Rate | Memory Usage (%) | Accuracy |
|---|---|---|---|---|
| Full KV Cache (Baseline) | 1.0x | 100% | 100% | 53.0 |
| SnapKV (Token Sel. only) | 1.0x | 50% | 50% | 52.66 |
| OjaKV (Rank Comp. only) | 1.67x (0.6x) | 100% | 60% | 43.13 |
| **OjaKV + SnapKV** | **1.67x (0.6x)** | **50%** | **30%** | **43.33** |

**Experimental Setup.** We chose SnapKV as it is a strong baseline that uses importance scores to identify and retain salient tokens. We evaluated four configurations on the LongBench benchmark suite using the Llama-3.1-8B model. The configurations are: (1) the uncompressed baseline, (2) SnapKV alone with a 50% token keep rate, (3) OjaKV alone with a 0.6x rank compression, and (4) a combined approach applying both OjaKV's rank compression and SnapKV's token eviction. Per-

formance is measured by the average accuracy across LongBench tasks, and efficiency is measured by the total KV cache compression ratio.

**Results and Analysis.** The results, presented in Table 6, confirm our hypothesis. Our analysis shows that **OjaKV** can be effectively combined with token eviction methods like SnapKV. This compounded approach further reduces KV cache memory usage with only a minor, graceful degradation in model accuracy. This result validates that our feature-dimension compression is complementary to sequence-length compression, offering a practical path to even greater memory efficiency.

