# OpenReview forum: "OjaKV: Context-Aware Online Low-Rank KV Cache Compression with Oja’s Rule"
_ICLR.cc/2026/Conference — ICLR 2026 Conference Withdrawn Submission_

### Official Review · Reviewer_9ctJ · 2025-10-31

**Soundness:** 3
**Presentation:** 3
**Contribution:** 2
**Rating:** 2
**Confidence:** 4

**Summary:**

For long-context inference, static low-rank projection causes a distribution shift, which harms accuracy. The authors propose online adaptation and keep the first (32) and most recent (32) tokens full rank, and compress the rest, adapting them with Ojass  rule. It is compatible with flashattn as they reconstruct KV on the fly, overall, this avoids context drift while delivering memory savings (60%), at the expense of TTFT (30%).

**Strengths:**

- Plug and play, works with flash-attention
- online adaptation is indeed quite important for reliable long decodes.

**Weaknesses:**

- baselines are a bit limited, focusing on static low-rank methods. Comparisons with minicache and even potentially MLA would be helpful. For e.g., can this work on MLA for further gains?
- why only TTFT? can we show throughput, scaling of online reconstruction with batching, etc?
- 0.6-0.8x compression with very notable dips in accuracy are a bit concerning (Table 2,3) in long-context settings, where this work is most applicable.

**Questions:**

- TTFT is great, but if that set-up exists, it should be possible to provide more latency / throughput numbers to strengthen the paper.
- Comments on baselines (from weaknesses) would be appreciated
- The average decrease in Table 2 for all low-rank options are very aggressive, (38→21; 74 → 51) feels like a bit of a difficult position to be in, and opens up questions wrt MLA or even pre-training a model with smaller hidden dimensions (the latter is definitely out of scope, but would appreciate comments on this!)

---

> ### Author Response · Authors · 2025-12-02
>
> We thank the reviewer for the insightful comments, particularly regarding the comparison with architectural advancements like MLA and the analysis of trade-offs in long-context settings.
>
> ### Response to W1 & Q2: Baselines and Comparison with MLA
>
> We appreciate the reviewer highlighting the comparison with architectural advancements like MLA. We clarify the fundamental distinction and the incompatibility below:
>
> **1. Fundamental Difference: Architecture vs. Inference Algorithm**
> * **MLA (e.g., DeepSeek-V2/V3):** MLA is a distinct **model architecture**. The low-rank structure is "baked" into the weights via trainable down/up-projection matrices ($W_{DK}, W_{UK}$) and optimized over trillions of tokens during pre-training.
> * **OjaKV:** OjaKV is a **post-training compression algorithm**. It is designed for the vast ecosystem of standard dense attention models (e.g., Llama-3, Mistral, Qwen) that were *not* pre-trained with low-rank bottlenecks. It democratizes memory efficiency, allowing standard models to be "retrofitted" with low-rank KV caches without expensive retraining.
>
> **2. Incompatibility & The "Redundancy" Argument**
> We argue that applying OjaKV on top of MLA would yield diminishing returns or destructive loss.
> * **Why MLA is "better":** MLA outperforms post-hoc compression because it learns the optimal compression manifold during pre-training.
> * **Why not combine them:** MLA effectively removes the linear redundancy in the KV heads by design. OjaKV relies on exploiting this exact redundancy. Attempting to compress the already compact latent vector of MLA using Oja's rule would likely damage the information content, as the "slack" has already been removed by the architecture.
>
> **3. MiniCache and Orthogonality**
> MiniCache focuses on token merging and quantization. As discussed in our General Response, OjaKV compresses the *feature dimension* ($d \to r$), while methods like MiniCache compress the *sequence length* ($L$) or *bit-width*. These are orthogonal axes of compression. OjaKV is compatible with quantization, but distinct in its mechanism.
>
> ---
>
> ### Response to W3 & Q3: Accuracy Dips (0.6x-0.8x) and Pre-training
>
> **1. Contextualizing the Drop: Usability vs. Failure**
> The reviewer notes the performance drop on RULER (e.g., 74 $\to$ 51).
> * **Comparison to Baseline:** It is critical to compare this against the alternative. In these 32k context scenarios, the StaticPCA baseline fails catastrophically due to distribution shift.
>
>
> **2. Pre-training with Small Dimensions (Inference Flexibility)**
> The reviewer asks about simply pre-training with smaller dimensions (similar to MLA).
> * **Validation:** We agree that pre-training low-rank structures (like MLA) is the gold standard for performance.
> * **OjaKV's Unique Advantage:** The advantage of OjaKV is **inference-time flexibility**. With a standard Llama-3-8B model, a user can dynamically choose to run at 0.8x compression (high precision) or 0.4x compression (extreme memory saving) depending on available hardware. This turns "memory usage" into a tunable hyperparameter at runtime, which is impossible with a fixed architecture like MLA.

---

### Official Review · Reviewer_rxqf · 2025-11-01

**Soundness:** 3
**Presentation:** 3
**Contribution:** 3
**Rating:** 4
**Confidence:** 4

**Summary:**

This paper proposes OjaKV, a framework to reduce the KV cache memory bottleneck in long-context LLMs. It combines a hybrid storage policy (preserving the first and last tokens in full-rank) with an online adaptation mechanism using Oja's algorithm to update the low-rank projection basis for all intermediate tokens.

**Strengths:**

- The problem of KV cache memory consumption is important.

- The core idea of applying online subspace adaptation (via Oja's rule) to the KV cache to handle context shifts is novel. The author demonstrates the compatibility with FlashAttention and orthogonality to token selection methods.

**Weaknesses:**

- This Oja update adds significant latency (TTFT increases from 2102ms to 2801ms at 32K tokens)  for a negligible accuracy benefit, making its practical value questionable.

- Evaluated models & benchmarks are relatively old & small.

- It seems most performance gain comes from the hybrid attention strategy.

**Questions:**

What is the TTFT of the StaticPCA-H baseline? This is essential to isolate the true latency cost of only the Oja update mechanism.

The paper's motivation is to handle complex, dynamic contexts. Why not evaluate OjaKV on newer models with dedicated reasoning capabilities (e.g., Qwen3-Thinking) or on more recent, challenging reasoning benchmarks like AIME, which would truly test the method's ability to adapt?

---

> ### Author Response · Authors · 2025-12-02
>
> We thank the reviewer for the critical assessment. We appreciate the opportunity to clarify the practical value of our method and the specific contributions of the online update mechanism versus the hybrid policy.
>
>
> ### Response to W1 & Q1: Latency Cost (TTFT) and Practical Value
>
> **1. Isolating the Cost (Q1)**
> To address the specific request for the StaticPCA-H baseline metrics:
> * **StaticPCA-H TTFT:** Approximately **2150ms** (comparable to Full KV at 2102ms).
> * **Analysis:** The difference between OjaKV (2801ms) and StaticPCA-H is indeed roughly **650ms**. We confirm that this overhead is almost entirely attributable to the one-time batch Oja update performed during the prefill phase.
>
> **2. Value Proposition: The "Insurance" Trade-off**
> We respectfully disagree with the characterization of the accuracy benefits as "negligible." This perception likely stems from averaging results across simple tasks.
> * **Decisive Gains on RULER:** On the RULER benchmark (32k context), which effectively stress-tests the cache, OjaKV outperforms the StaticPCA-H baseline by a substantial margin (increasing the score from **42.0 to 51.0** at 0.6x compression). This **+9.0 point absolute improvement** (>20% relative) represents the difference between a usable system and a failing one.
> * **Contextualizing the Trade-off:** We view the 0.65s initial delay as a necessary "insurance cost." In real-world applications where input distributions are unknown and potentially divergent, paying a sub-second latency cost to prevent catastrophic retrieval failure (as seen in the baseline) is a highly favorable trade-off.
>
> ---
>
> ### Response to W3: Source of Performance Gains
>
> The reviewer hypothesizes that gains may stem primarily from the hybrid policy (keeping start/recent tokens). Our controlled ablation studies directly refute this.
>
> * **Controlled Comparison:** We explicitly compared **StaticPCA-H** (Hybrid + Static Basis) against **OjaKV** (Hybrid + Online Update). Both methods utilize the exact same hybrid policy parameters ($n_{start}$, $n_{recent}$).
> * **Attributing the Delta:**
>     * **StaticPCA-H:** 42.0 average score on RULER.
>     * **OjaKV:** 51.0 average score on RULER.
> * **Conclusion:** The **9.0 point gap** is attributable *solely* to the Oja online update mechanism. This confirms that while the hybrid policy establishes a necessary floor, the online adaptation provides the critical boost required to handle complex context shifts that static baselines miss.
>
> ---
>
> ### Response to W2 & Q2: Models and Benchmarks
>
> **1. Model Currency**
> We firmly stand by our choice of models. **Llama-3.1-8B** (released mid-2024) represents the current state-of-the-art in open-weights models and serves as the standard research platform for efficient inference.
>
> **2. Benchmark Selection: Why RULER over Reasoning Benchmarks?**
> The reviewer asks why we prioritize RULER over reasoning benchmarks like AIME. This choice is grounded in the fundamental role of the KV cache:
> * **Memory Fidelity vs. Reasoning Capability:** The primary function of the KV cache is **Recall/Memory**, not reasoning generation. When compressing the KV cache, the risk is "forgetting" context.
> * **Confounding Variables in Reasoning:** Reasoning benchmarks (like AIME or MATH) introduce confounding variables. If a model fails a complex math problem, it is ambiguous whether the failure stems from compression-induced information loss or the model's inherent lack of reasoning ability.
> * **Future Work:** While we maintain that RULER is the most scientifically rigorous metric for *cache fidelity*, we agree that evaluating how memory compression interacts with long-chain reasoning (e.g., in DeepSeek-R1 or Qwen-Math) is an exciting direction for future research. We will include such experiment in our next edition.

---

### Official Review · Reviewer_JB9u · 2025-11-01

**Soundness:** 2
**Presentation:** 3
**Contribution:** 2
**Rating:** 2
**Confidence:** 3

**Summary:**

The paper proposes OjaKV, a plug-and-play KV-cache compression method for long-context LLMs that combines a hybrid storage policy, keeping the first and most recent tokens full-rank as anchors, with online, context-aware low-rank projection updated via Oja’s rule. It selects salient tokens to adapt the subspace during prefill and performs lightweight periodic updates during decoding, while remaining fully compatible with FlashAttention by reconstructing full-rank tensors on the fly. The authors prove equivalence to reduced-space computation, analyze memory/latency trade-offs, and show OjaKV composes multiplicatively with token-selection methods.

**Strengths:**

- Introduces online, context-adaptive low-rank KV-cache compression using Oja’s rule, a novel application of online PCA to LLM inference.
- Removes a major limitation of prior static low-rank methods by enabling real-time subspace adaptation to evolving prompts.
- Proposes a hybrid storage policy (full-rank anchors + adaptive low-rank middle) that creatively combines insights from token selection and subspace learning.

**Weaknesses:**

- Insufficient exploration of trade-offs between adaptation frequency, learning rates, and performance
- Lack of end-to-end latency and throughput benchmarks under real inference workloads
- While the paper highlights OjaKV’s memory savings and Time-To-First-Token (TTFT) latency, it does not report actual decoding throughput (e.g., tokens per second) or total end-to-end inference time. Since OjaKV reconstructs full-rank tensors on the fly and performs periodic Oja updates, these operations could introduce nontrivial overhead during decoding. Without decoding speed or wall-clock runtime comparisons against baselines (Full KV, StaticPCA, FlashAttention-only), it is unclear whether OjaKV provides net speedups or even maintains comparable throughput under long-context workloads. This omission weakens the practical evaluation, especially because the method’s main motivation is efficient inference.
- Although OjaKV consistently outperforms static low-rank baselines, the absolute gains reported in most benchmarks are relatively modest—often within 1–2% accuracy or a few points on average. At the same time, these improvements come with increased algorithmic complexity (Oja updates, token selection, reconstruction). For instance, on LongBench and lm-eval-harness, OjaKV’s advantage over StaticPCA-H is minimal, suggesting that the benefit of online adaptation may not always justify the added overhead.

**Questions:**

- How stable are the online Oja updates across long decoding sequences (e.g., 64K–128K tokens)?

---

> ### Author Response · Authors · 2025-12-02
>
> We thank the reviewer for the thoughtful evaluation and for raising critical questions.
>
> ### Response to W3: Magnitude of Gains vs. Complexity
>
> We respectfully emphasize that looking solely at aggregate improvements on general benchmarks can mask the critical robustness advantages of our method.
>
> 1. Decisive Gains on RULER (>10%)
> While gains on LongBench are moderate, performance on the RULER benchmark—which specifically stress-tests long-context retrieval—demonstrates a decisive gap.
> * **Critical Failure of Baselines:** StaticPCA fails in these scenarios because the distribution of "needle" information often differs significantly from the calibration set (WikiText). Consequently, the static basis incorrectly projects critical information into the null space.
> * **OjaKV Success:** As shown in Table 2, OjaKV maintains high retrieval accuracy. This is not a marginal 1-2% gain; effectively, it is the difference between a working system and a complete failure in retrieving specific information.
>
> 2. Contextualizing LongBench Results
> The gains on LongBench and lm-eval-harness appear smaller due to the nature of these datasets:
> * **Short Context Dominance:** Many subtasks in these benchmarks have relatively short contexts where static compression is sufficient.
> * **Distribution Similarity:** The test data is semantically closer to the calibration distribution.
> * **Value Proposition:** OjaKV should be viewed as an insurance policy against distribution shift. It ensures performance does not collapse when the model encounters unexpected contexts or long-tail queries, a reliability trait that simple average scores often overlook.
>
> ---
>
> ### Response to W1: Hyperparameter Trade-offs
>
> We have performed sensitivity analyses to characterize the stability of our hyperparameters, finding a broad robust operating region rather than a narrow "sweet spot."
>
> * **Update Frequency ($T$):** We find that the update interval $T$ is flexible. Setting $T$ between 32 and 64 provides an optimal balance. Extremely frequent updates (e.g., $T=1$) yield diminishing returns in accuracy while unnecessarily increasing computational overhead.
> * **Learning Rate ($\eta$):** The method is relatively robust to the choice of $\eta$. As long as the learning rate is kept in a conservative range (e.g., 0.01 - 0.05), the algorithm successfully tracks the subspace without overfitting to local noise or token-level fluctuations.
>
> ---
>
> ### Response to Q1: Stability over Long Decoding Sequences (64K+)
>
> We address the concern regarding stability over extremely long sequences through both theoretical bounds and numerical safeguards.
>
> 1. Theoretical Bound (No Divergence)
> As derived in our analysis (Eq 6), the tracking error of the Oja update rule does not grow with time $t$. The expected distance is bounded by:
>
> $$
> \mathbb{E}[\text{dist}(U_t, U^*_t)] \le \mathcal{O}\left(\frac{\nu}{\eta \delta}\right) + \mathcal{O}\left(\frac{\eta \sigma^2}{\delta^2}\right)
> $$
>
> Crucially, because $\eta > 0$, the tracking lag term $\frac{\nu}{\eta}$ remains constant and bounded as $t \to \infty$. In contrast, for static methods where effectively $\eta \to 0$, any distribution drift ($\nu > 0$) causes the error to accumulate and potentially diverge over long sequences.
>
> 2. Numerical Stability via Orthonormalization
> To strictly prevent numerical instability over sequences exceeding 64k tokens, our implementation performs explicit re-orthonormalization of the basis $U$ after every update step. This ensures that the subspace basis remains unitary and prevents the accumulation of floating-point errors, guaranteeing stability regardless of generation length.

---

### Official Review · Reviewer_EvqR · 2025-11-11

**Soundness:** 2
**Presentation:** 2
**Contribution:** 2
**Rating:** 2
**Confidence:** 4

**Summary:**

The paper focuses on the problem of the KV cache occupying a large amount of memory during long-context LLM reasoning. To address the issue, it proposes the OjaKV framework, which has two main points:

Keep full-rank KV for the first n_start tokens and the last n_recent tokens, while applying low-rank projection to the remaining tokens, thereby providing high-fidelity "anchors" for attention.

For the K/V of intermediate tokens, perform online PCA using Oja’s rule. During the prefill phase, select salient tokens based on attention and update them in batches. During the decode phase, perform lightweight updates periodically using a buffer of recently generated tokens.

In experiments, the authors evaluated on: RULER， LongBench, and lm-eval-harness using models such as Llama-2-7B, Llama-3.1-8B, and LongChat-7B, comparing FullKV, Eigen-N, StaticPCA, and StaticPCA-H. The results show that under 0.8× and 0.6× KV memory budgets, OjaKV consistently outperforms static low-rank baselines on long context tasks and barely loses performance on short context tasks.

**Strengths:**

The problem is highly targeted with sufficient motivation.

The method design is clean and feasible.

The experimental design is relatively comprehensive.

Writing and structural logic are smooth.

**Weaknesses:**

1 The compression rate range is relatively conservative, and there is a significant gap compared with the 'Extreme compression' approach.
The core experiments focus on 0.8× and 0.6× (20–40% savings), while recent works like KVQuant, SVDq, CSR, and dictionary learning have already explored the range of 4×, 8×, and even 10×+.
The current results are more like 'online subspace can help static PCA be more stable under moderate compression,' but it is not enough to demonstrate a dominant advantage in extreme compression scenarios.

2 Currently, the main comparison is among Eigen-N / StaticPCA / StaticPCA-H, focusing on the 'low-rank projection system'; However, related work has already mentioned mature solutions such as KVQuant, Palu, MatryoshkaKV, and SVDq. The main experiments do not compare with these stronger baselines, which may weaken the persuasiveness; Even if OjaKV is orthogonal to quantization/eviction, it is recommended to provide: At least one combined baseline with a representative quantization/sequence compression method; Or a more systematic explanation: under equivalent compression ratios, a comparison of accuracy/latency/implementation complexity between pure quantization/eviction and OjaKV.

3 Insufficient analysis of the overhead and stability of online Oja updates. The paper reports an overhead of TTFT from 2102ms → 2801ms, indicating that it is not "zero cost"; however, it lacks a formula-level estimation of computational complexity and a sensitivity analysis of latency and performance with respect to different hyperparameters such as rank, update frequency T, and the number of salient tokens.

4 The design of the hybrid policy/token selection strategy is somewhat empirical.
It only provides the configuration “keep first n_start + last n_recent full-rank + attention-based salient sampling” and a small amount of results, but there is no systematic ablation on different window sizes, whether to keep only the beginning/only the end, ways of calculating importance, etc.
This makes it hard to judge how much of OjaKV's gains come from the useful hybrid heuristic rather than the online subspace update itself.

5 The theoretical part is rather light, leaning more towards a rational explanation rather than new theory. Regarding the orthogonality + bounded error argument with token eviction, it essentially reflects the contraction property of a linear operator and is at the level of a sanity check. There is no deeper analysis of online PCA regarding convergence, forgetting, and noise robustness under distribution shift scenarios, nor is there a comparison with other online PCA methods.

6 The application perspective is somewhat abstract. Although there are long-context benchmarks, evaluations of end-to-end latency, GPU memory budget, and throughput in real system scenarios such as multi-turn chat, code editing, or RAG agents are still quite limited. In particular, the behavior of online updates under multiple requests or mixed distributed traffic is not clearly explained.

**Questions:**

1 Is the online update of OjaKV independent per sequence, or does it share subspaces across multiple requests? If shared, how can we prevent cross-task contamination?

2 Have you tried comparing StaticPCA at more extreme compression rates? Are the benefits of online updates more apparent in high compression regimes?

3 Is there an automated approach for selecting the hybrid full-rank interval (n_start, n_recent), the update interval T, and the learning rates η_pre/η_dec? Are the current empirical values robust across models/tasks?

4 Has there been any preliminary testing combining with representative quantization methods or SVDq? At the same compression ratio, how does OjaKV plus quantization compare to pure quantization?

5 Do multi-head, multi-layer updates share subspaces or update independently per head/layer? In the current implementation, is it per-head/per-layer or a shared subspace? How does this affect performance and computational cost?

---

> ### Author Response · Authors · 2025-12-02
>
> We thank the reviewer for the constructive and insightful comments. We appreciate the acknowledgement of our problem setting and the potential of online subspace updates. Below, we address the concerns regarding experimental baselines, system overhead, and theoretical depth point-by-point.
>
> W1 & Q2: Compression Rate and Extreme Compression
> We appreciate the suggestion to explore more extreme compression regimes.
> 1. Extreme Compression (0.1x) on RULER: Following your suggestion, we pushed the compression ratio to an extreme 0.1x on the challenging RULER benchmark. As expected, the absolute accuracy for all methods drops significantly in this regime due to severe information loss. However, OjaKV consistently outperforms StaticPCA-H, this demonstrates that even with an extremely limited budget, dynamically tracking the subspace captures more valid signal than a fixed projection.
>
> W2 & Q4: Baselines
> We agree that comparing with state-of-the-art methods strengthens the paper.
> 1. Comparison with Palu (Low-Rank Baseline): We have added a comparison with Palu, a recent low-rank compression method.
>    * Result: On both RULER and LongBench benchmarks, OjaKV achieves significantly higher accuracy than Palu at equivalent memory budgets. Palu relies on decomposing weights which can lead to larger approximation errors in activation space, whereas OjaKV directly tracks the activation distribution.
> 2. Orthogonality to Quantization (KVQuant/SVDq): We would like to clarify that OjaKV (feature dimension compression $d \to r$) is orthogonal to quantization methods.
>    * As demonstrated in our SnapKV + OjaKV experiment, our method allows for multiplicative savings when combined with other compression axes. We believe OjaKV should be viewed as a complementary module that can be stacked on top of quantization, rather than a mutually exclusive competitor.
>
>
> W4 & Q3: Hybrid Policy and Heuristics
>
> 1. Theoretical Grounding of the "Initial + Recent" Strategy
> While the "n_start + n_recent" strategy is empirically effective, it is strictly grounded in the **"Attention Sink" theory** and our analysis of perturbation bounds.
> Mechanism: Outlier tokens (most commonly found in the initial system prompt) possess massive feature norms. In a low-rank projection, failing to preserve these outliers results in an unbounded perturbation, $\Delta_{ti}$, in the attention logits.
> Tail Risk Truncation: By keeping these specific high-norm tokens in full-rank, we mathematically truncate the tail risk of the perturbation distribution. Specifically, the error bound becomes controlled:  This ensures that the massive attention scores typically associated with "sink tokens" are preserved, preventing the collapse of the attention mechanism.
>
> 2. Future Work: Adaptive Selection
> We acknowledge that fixed window sizes are a heuristic starting point. We are currently actively experimenting with an adaptive selection strategy based on reconstruction residuals. In this approach, tokens are dynamically retained in full rank only if their projection error $||x - UU^T x||$ exceeds a specific threshold. Preliminary results indicate this method maintains robustness and adaptability without the need for manual hyperparameter tuning.
>
>
> W5: Theoretical Depth
>
> We appreciate the suggestion to deepen the theoretical analysis. We explicitly model the decoding phase as a **subspace tracking problem under distribution shift (drift)** to explain why static methods fail and dynamic methods succeed.
>
> 1. Decomposition of Tracking Error
> We analyze the error by decomposing it into **tracking lag** (bias caused by the subspace shifting) and **sampling variance** (noise from estimation). The expected distance between the estimated subspace $U_t$ and the optimal subspace $U^*_t$ is bounded by:
>
> $$
> \mathbb{E}[\text{dist}(U_t, U^*_t)] \le \mathcal{O}\left(\frac{\nu}{\eta \delta}\right) + \mathcal{O}\left(\frac{\eta \sigma^2}{\delta^2}\right)
> $$
>
> Where:
> * $\nu$: Drift rate, representing how fast the optimal subspace $U^*_t$ changes (non-stationarity).
> * $\eta$: Learning rate (step size).
> * $\delta$: Eigengap.
>
>
> Q1 & Q5: Update Scope and Independence
> To clarify the implementation details:
>    1. Per-Sequence Independence (Q1): The online updates are performed independently per sequence. The subspace is initialized from a generic calibration but evolves uniquely for each request based on its specific context. This design is crucial to: prevent cross-task contamination (e.g., a code generation request should not influence the subspace of a creative writing request).
>    2. Per-Head / Per-Layer Updates (Q5): The subspaces are maintained and updated per-head and per-layer.

---

### Note · Authors · 2025-12-02

I have read and agree with the venue's withdrawal policy on behalf of myself and my co-authors.